**Global patterns of soil organic carbon distribution in the 20－100 cm soil profile**
**for different ecosystems: A global meta-analysis**
Haiyan Wang[1,2], Tingyao Cai[1], Xingshuai Tian[1], Zhong Chen[1], Kai He[1], Zihan Wang[1],
Haiqing Gong[1], Qi Miao[1], Yingcheng Wang[1], Yiyan Chu[1], Qingsong Zhang[1], Minghao
Zhuang[1], Yulong Yin[1, ]*, Zhenling Cui[1]
[1] State Key Laboratory of Nutrient Use and Management, College of Resources and
Environmental Sciences, China Agricultural University, 100193 Beijing, China
[2] Sanya Institute of China Agricultural University, 572025 Sanya, China
*Corresponding author**: Yulong Yin. Email: yinyulong88221@163.com
**Manuscript submitted to: Earth System Science Data**
**Abstract**
Determining the distribution of soil organic carbon (SOC) in subsoil (20–100 cm depth)
is important with respect to the global C cycle and warming mitigation. However,
significant knowledge gaps remain regarding the spatiotemporal dynamics of SOC
within this layer. By integrating traditional depth functions with machine learning
approaches, we quantified soil β values, which represent the relative rate of decline in
SOC density with depth, and provided high-resolution assessments of SOC dynamics
across global ecosystems, including cropland, grassland, and forestland. The estimated
subsoil SOC densities were 62 Mg ha$^{-1}$ (95% CI: 52-73) for cropland, 70 Mg ha$^{-1}$ (95%
CI: 57-83) for grassland, and 97 Mg ha$^{-1}$ (95% CI: 80-117) for forestland. SOC density
exhibited a consistent decline with depth, ranging from 30 Mg ha$^{-1}$ to 5 Mg ha$^{-1}$ in
cropland, 32 Mg ha$^{-1}$ to 7 Mg ha$^{-1}$ in grassland, and 40 Mg ha$^{-1}$ to 13 Mg ha$^{-1}$ in
forestland, across 20 cm depth increments from 20 to 100 cm. The estimated global
subsoil SOC stock was 803 Pg C, with cropland, grassland, and forestland contributing
74 Pg C, 181 Pg C, and 547 Pg C, respectively. On average, 57% of this carbon was
stored within the top 0-100 cm of the soil profile. This study provides information on
the vertical distribution and spatial patterns of SOC density at a 10 km resolution across
global ecosystems, providing a scientific basis for future studies pertaining to Earth
system  models.  The  dataset  is  open-access  and  available
at https://doi.org/10.5281/zenodo.15019078 (Wang et al., 2025).
**Keyword:** Subsoil SOC distribution; Soil profiles; Random Forest; Driving factors;
Global ecosystems

## 1. Introduction

Soil organic carbon (SOC) plays a pivotal role in global carbon cycling, climate change mitigation, reducing greenhouse gas emissions, while simultaneously supporting ecosystem health (Bradford et al., 2016; Lal et al., 2021; Griscom et al., 2017). Subsoil, defined here as the soil layer below 20 cm, contains over half of the global SOC stock (Jobbágy & Jackson, 2000; Poffenbarger et al., 2020; Batjes, 1996). However, the extensive loss of SOC through agricultural practices such as crop production and grazing has substantially contributed to rising atmospheric $CO_2$ levels (Beillouin et al., 2023; Lal, 2020; Qin et al., 2023). Complex polymeric carbon in subsoil is vulnerable to decomposition under future warming. Specifically, ecological or trophic limitations of SOC biodegradation in deep soil layers can lead to sharp declines in the nutrient supply and biodiversity (Chen et al., 2023). Subsoil is better suited to long-term C sequestration than topsoil. The '4 per 1000' initiative aims to boost SOC storage in agricultural soils by 0.4% annually, offering a potential pathway for mitigate climate change and increase food security (Chabbi et al., 2017). Promoting subsoil carbon sequestration, particularly in agricultural and managed ecosystems, could facilitate the long-term stabilization of fossil-fuel-derived carbon in soils (Button et al., 2022). Despite the importance of subsoil organic carbon dynamics, we were still poorly understood, especially at large scale (Padarian et al., 2022). This is primarily due to the challenges associated with measuring SOC at greater depths, which is difficult, time-consuming and labor-intensive.

Recent studies have focused on SOC allocation and dynamics at varied depths and the subsoil SOC–Climate feedback cycle of terrestrial ecosystems (Luo et al., 2019; Jia et al., 2019; Li et al., 2020). The complexity, uncertainty, and large spatial heterogeneity of SOC stock estimation have limited the ability to accurately quantify the SOC stock distribution (Mishra et al., 2021; Wang et al., 2022a). Currently, three primary methods are commonly used to estimate large-scale SOC stocks: (1) area-weighted averaging based on vegetation inventories and soil survey data (Tang et al., 2018); (2) machine-learning based on remote-sensing, land-use, and edaphic data and climatic factors as covariates (Ding et al., 2016); and (3) depth distribution function-based empirical analysis (Wang et al., 2023). The first approach provides the most accurate measurement of the SOC stock, but is time-consuming and labor intensive and is not practical at the global scale. The latter two do not fully consider the vertical distribution

of the soil profile or the soil properties of various ecosystems. Extrapolating surface
SOC measurements from 0–40 cm or 0–50 cm to predict subsoil SOC at greater depths,
such as 0–100 cm or 0–200 cm, introduces significant uncertainty, hindering precise
estimation of the global subsoil SOC stock (Wang et al., 2023; Ding et al., 2016).
Studies of whole-soil profiles have recorded greater changes in the SOC dynamics of
the subsoil under warming (Zosso et al., 2023; Luo et al., 2020; Soong et al., 2021).
The amount and quality of C in input soil, such as aboveground litter and root biomass
input, could profoundly alter the vertical SOC distribution (Lange et al., 2023; Feng et
al., 2022). The β model, in particular, uses simple and flexible functions that capture
the relative slope of depth profiles with a single parameter, with the advantage of being
able to integrate SOC values from the surface down to a given depth (Jobbágy and
Jackson., 2000). The β model was originally applied to vertical root distributions and
has been used to fit the steepest reductions with depth (Gale and Grigal, 1987; Jackson
et al., 1997). Some researchers have used the global average β of 0.9786 to calculate
deep soil SOC stocks (Yang et al., 2011; Deng et al., 2014). However, the different
hydrological conditions, soil type, and ground/underground organic matter have limited
the ability to resolve the SOC depth distribution with confidence.
In this study, we produced spatially resolved global estimates of the depth distribution
and stocks of subsoil SOC using the β model as a depth distribution function-based
empirical approach for evaluating cropland, grassland, and forestland ecosystems on a
global scale. We collected and analyzed 17,984 observation data from globally
distributed soil profiles (0–100 cm) across 14,550 sites to estimate soil β values. We
then developed a random forest (RF) model to estimate the spatial variation in grid-
level soil β values in the associated ecosystems to resolve the dynamics of the SOC
density in different soil layers and subsoil stocks of the global ecosystems.

## 2.Methods

### 2.1. Data collection

We conducted peer-reviewed literatures review of studies previously published on SOC
stock or SOC content of soil profile between 1980 and 2023 to obtain a database. The
Web of Science and China National Knowledge Infrastructure (CNKI) database were
searched using the terms "Soil organic carbon" AND "Soil profile" OR "Subsoil" OR
"Deep soil". And the criteria were as follows: (1) The research scope is worldwide. (2)

The study was conducted in the field. (3) The profiles of multiple sites are reported in the same literature, and the profile of each site is considered as an independent study. (4) Profiles with more than three suitable measurements of organic carbon in the first meter were collected from the analysis for there was sufficient detail to characterize the vertical distribution of SOC. (5) The data extracted from included basic site information including location latitude and longitude, soil organic carbon (SOC), total nitrogen (TN), soil bulk density (BD), soil pH and CN ratio, Microbial biomass carbon and nitrogen (MC), Microbial biomass nitrogen (MN), soil clay content, climate conditions (mean annual precipitation (MAP) and mean annual temperature (MAT)). If the soil organic matter (SOM) rather than SOC was reported, the value was converted to SOC by multiplication with a conversion factor of 0.58 (Don et al., 2011). To extract data presented graphically, the digital software GetData Graph Digitizer 2.25 (getdata-graph-digitizer.com) was used. A total of 209 peer-reviewed papers comprising 1,221 soil profiles were included in this dataset, including 758 for cropland, 219 for forestland, and 244 for grassland. Additionally, an expanded dataset was sourced from the WoSIS Soil Profile Database, contributing 7,636 profiles for cropland, 4,534 for forestland, and 4,593 for grassland (Figure 1a). Missing soil and climate factor data from a few sites were either provided by the study authors through direct correspondence, or obtained from the spatial datasets (section 2.2), based on latitude and longitude. These completed data were analyzed to determine the impact of the environment on soil β values and develop a model to predict global grid-level β values, subsequently, soil profiles SOC density, and calculate SOC stocks. Additionally, the soil samples are classified into four major types: sandy soil, loam, clay loam, and clay soil, according to the international soil texture classification standard (Zhao et al., 2022).

### *2.2 Calculation of soil attributes from literature-derived database*

Since the 0–1 m soil profile has different layers in the row data, mass-preserving spline method (R Package 'mpspline2') was used to divide the soil profiles into 5 layers with 20 cm interval. This function implements for continuous down-profile estimates of soil attributes (SOC, TN, Clay, MC, MN, etc.) measured over discrete, often discontinuous depth intervals. In some studies, bulk density data below the 20 cm soil layer were lacking. Notable differences in global SOC stocks estimations were attributed to the values used for soil bulk density. Therefore, we use the database issued by predecessors to generate bulk density data with 0-1m profile at 20 cm interval (Shangguan et al.,

2014). The equation used to calculate SOC density at each research site was the following:

$$SOC\ density = SOC * BD * D * (1 - GC/100)/10 \qquad [1]$$

where SOC is the SOC concentration (g kg$^{-1}$), BD is the soil bulk density (g cm$^{-3}$), and D is the thickness of the soil layer (at intervals of 20 cm in the first meter), SOC density (Mg C ha$^{-1}$). GC (>2 mm) is the gravel content (%).

### 2.3 Calculation of soil β values from literature-derived database

To enhance the comparability of data from different studies, the corresponding soil β values were calculated using Equation 2, which follows the methodology adopted by Yang et al. (2011). The SOC density in the top 0–100 cm was calculated from the initial depth SOC density using Equation 3, which was developed by Jobbágy & Jackson (2000). The equations are as follows:

$$Y = 1 - \beta^d \qquad [2]$$

$$X_{100} = \frac{1 - \beta^{100}}{1 - \beta^{d_0}} * X_{d0} \qquad [3]$$

where Y represents the cumulative proportion of the SOC density from the soil surface to depth d (cm); β is the relative rate of decrease in the SOC density with soil depth; A lower β indicates a steeper decline with depth. $X_{100}$ denotes the SOC density within the upper 100 cm; $d_0$ represents the depth of the 0-20 cm soil layer; (cm); and $X_{d0}$ is the SOC density of the top 20 cm soil depth.

### 2.4 Spatial gridded datasets

The gridded datasets included forestland, grassland, and cropland areas, climate factors and soil properties. Areas of cropland, forestland, and grassland were obtained from Global Agro-Ecological Zones (GAEZ, https://gaez.fao.org/) at a resolution at 0.083° × 0.083°. The MAP and MAT were acquired from the Climatic Research Unit Time Series (CRU TS ver. 4.05; (https://crudata.uea.ac.uk/cru/data/hrg/cru_ts_4.05/cruts.2103051243.v4.05/). The spatial SOC, total N, soil clay contents, and soil pH and gravel content were acquired from the Harmonized World Soil Database ver. 1.2 (https://www.fao.org/soils-portal/data-hub/soil-lassification/worldreference-base/en/). MC and MN data were obtained from this study (Xu et al., 2013). The BD and gravel content (GC) datasets of

the whole soil profile was acquired from Harmonized World Soils Database version 2.0
(HWSD v2.0) (https://gaez.fao.org/pages/hwsd), whose resolution is 1 km. The
belowground net primary productivity (BNPP) data were sourced from Xiao et al.
(2023). All data were resampled at 0.083° resolution using the "raster" R package
(https://rspatial.org/raster).
*2.5 Application of RF modeling to predict spatial β values*
We reconstruct the relationships among multiple factors, cropland, grassland and
forestland soil β values by RF algorithm. The developed RF models were used to predict
grid-level soil β values for each ecosystem. Prior to constructing the RF model, the
optimal parameter values of $m_{try}$ and *ntrees* were determined through the bootstrap
sampling method, which was performed with the "e1071" R package. Predictions of
soil β values derived by RF and random-effects regression models were evaluated by
10-fold cross-validation. The dataset was divided into 10 subsets of equal size, of which
70% were used for model fitting and RF procedures, then predicted with the fitted
models using the remaining 30% of the data. The performance of RF models was
evaluated based on the coefficient of determination ($R^2$) and root mean square error
(RMSE) according to those following equations:
$$R^2 = 1 - \frac{\sum_{p=1}^{q}(y_p - \hat{y}_p)^2}{\sum_{p=1}^{q}(y_p - \bar{y})^2} \tag{4}$$

$$RMSE = \sqrt{\frac{\sum_{p=1}^{q}(y_p - \hat{y}_p)^2}{q}} \tag{5}$$

where $y_p$ represents an observed value (p = 1, 2, 3, …), $\hat{y}_p$ represents the
corresponding predicted value (p = 1, 2, 3, …), $\bar{y}$ represents the mean value of
observed values, and q represents the total number of observed values.
*2.6 Estimating global SOC density and SOC stocks ecosystems across different*
*ecosystems*
To reveal the dynamics of SOC with depth, we used the globally predicted β values for
cropland, grassland, and forestland ecosystems in Equation 3 to calculate cumulative
SOC density at specific depths (e.g., 40, 60, 80, and 100 cm). Based on these cumulative
values, the SOC density for each 20 cm interval as calculated by subtracting the
cumulative SOC density of the shallower depth from the deeper depth. Subsequently,
the total carbon stocks for different ecosystems worldwide were calculated by
multiplying the SOC density by the corresponding land area (see Equation 6).

$$SOC\ stocks = SOC\ density * S_{ecosystem} \qquad [6]$$

Where $S_{ecosystem}$ is the areas of cropland, grassland or forestland (ha), SOC stocks (Pg C).

### 2.7 Uncertainty analysis

A Monte Carlo simulation was used to estimate the overall uncertainty in the estimated spatial SOC density. The uncertainty mainly came from be soil β estimation-related parameters and the RF model. Input parameters in the RF model prediction followed independent normal distributions by assuming the grid value as the mean value and its 10 % as the standard deviation (Liu et al., 2024; Xu et al., 2023; Vande et al., 2004). Then, 1,000 random samplings were used to obtain the interval of each grid via Monte Carlo simulations. The sampling value was then used to run the RF model to predict the grid-level soil β with 100 bootstraps to run the RF model. Then we used predicted grid-level soil β to recalculated the distribution of SOC density (SOCD) across different ecosystem. Finally, we calculated the mean along with the 2.5% and 97.5% percentiles to establish the 95% confidence interval of SOC density and SOC stocks.

$$U_i = \frac{CI_i}{x_i} \qquad [7]$$

Where $x_i$ is the mean of prediction, $CI_i$ is the confidence interval of $x_i$, $U_i$ is the uncertainty

### 2.8 Data management and analyses

One-way analysis of variance at $p < 0.05$ was applied to identify significant differences in soil β values using SPSS ver. 20.0 (SPSS, Inc., Chicago, IL, USA) software. We made a database of peer-reviewed publications with Excel 2010 software (Microsoft Corp., Redmond, WA, USA). Weather data analyses were performed using MATLAB R2017a software (MathWorks Inc., Natick, MA, USA). Weather data were analyzed using MATLAB R2017a (MathWorks, Natick, MA, USA). R software (ver. 3.5.1; R Development Core Team, Vienna, Austria) was used to generate graphs. A publicly available map of China was obtained from the Resource and Environment Data Cloud Platform (http://www.resdc.cn). All map-related operations were implemented using ArcGIS 10.2 software (http:/www.esri.com/en-us/arcgis). All algorithms implemented using the random Forest R package in the R software environment (ver. 3.5.1; R Development Core Team, Vienna, Austria).

### 3. Results

### 3.1 Soil β values of the three global ecosystems based on field measurements

We analyzed 17,984 globally distributed soil β values (calculated by SOC density and depths) from 14,550 sites, including 5,940 cropland, 4,209 grassland, and 4,401 forestland sites (Figure 1a). This included an additional 8,394 observations for cropland, 4,753 for forestland, and 4,837 for grassland, obtained from the literature and the WoSIS Soil Profile Database. The average soil β values across all observations were 0.9731 for cropland, 0.9772 for grassland, and 0.9790 for forestland (Figure 1b), with significant differences observed among the ecosystems. Soil β values exhibited significant differences among sandy soil, loam, clay loam, and clay soil. Cropland and grassland ecosystems exhibited the highest β values in sandy soil, while forest ecosystems showed the highest β values in clay soil (Figure 1c-d).

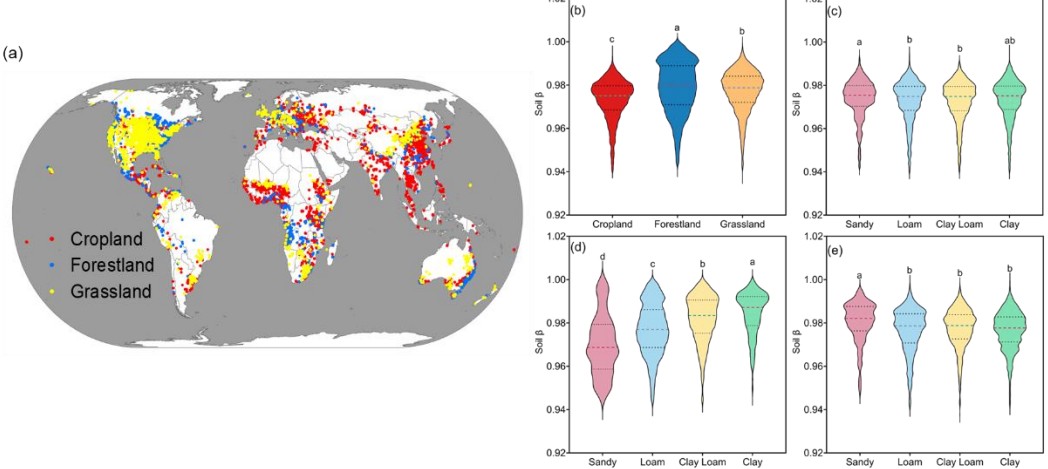

**Figure 1.** Geographic location of the study sites included in the meta-analysis of the 0–100 cm soil profiles (**a**). Red, yellow, and blue dots represent cropland, grassland, and forestland, respectively. Soil β values of the study sites showing significant differences in different ecosystems with ANOVAR analysis and Duncan's new multiple range test (**b**). **c-e** demonstrate the variations in soil β values across sandy soil, loam, clay loam, and clay for cropland, forestland, and grassland, respectively.

### 3.2 Impact of soil and climate variables on soil β values

The soil β value is significantly influenced by the combined effects of various climatic, biological, and soil factors. MAT, MAP and BNPP were the most influential driver of β values (Figure S1). Higher MAT promoted increases in soil β values and higher MAP promoted decreases; however, when the MAT was about 20°C and MAP was about

1000 mm, the soil β values growth and decline rate was substantially reduced (Figure
2a and b). BNPP demonstrated a nonlinear relationship: β values decreased with
increasing BNPP levels, when BNPP was below 1.5 Mg ha$^{-1}$ yr$^{-1}$ and exceed 2 Mg ha$^{-}$
$^{1}$ yr$^{-1}$, the soil β values decreased sharply (Figure 2c). The regression between CN, MC,
MN, TN, pH and soil β values was parabolic. When CN >10, MC >100 mg/kg, MN >20
mg/kg, TN >3 g/kg and pH <6, the soil β value promoted decreased (Figure 2d, e, f, g
and h). β values remained relatively stable across most clay percentages but showed a
decrease when clay content exceeded 30% (Figure 2i). Through comparison and
analysis, we ultimately selected 9 significant factors (BNPP, pH, Clay, MAT, MAP,
TN, MN, MC, CN) for modeling based on their importance and explanatory power
(Figure S1).

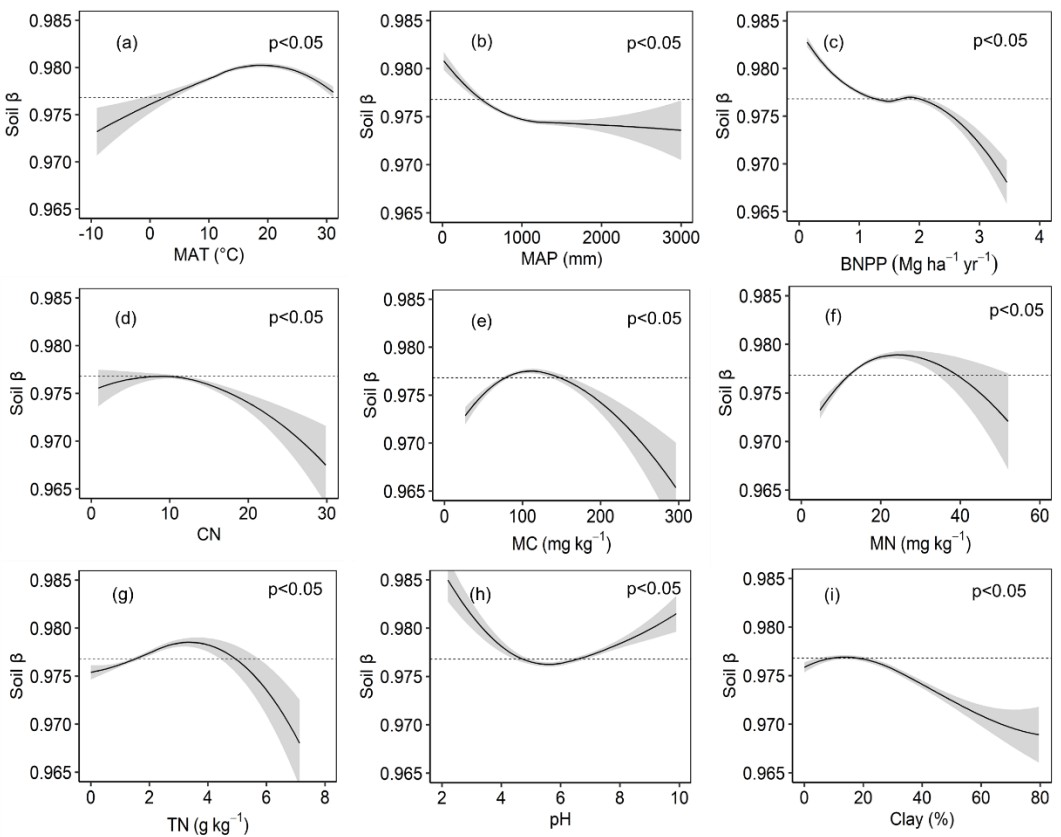


**Figure 2. a-i** show the variables affecting soil β values. MAT, mean annual temperature;
MAP, mean annual precipitation; BNPP, belowground net primary productivity; CN,
the ratio of SOC to TN; MC, microbial biomass carbon; MN, microbial biomass
nitrogen; TN, soil total nitrogen; pH, soil pH; Clay, clay content. Shaded bands indicate
95% confidence intervals, and the dashed lines represent the average soil β values.
*3.3 Performance of the random forest regression model*
We developed an RF regression model using machine learning techniques to determine
grid-level soil β values on a global scale. The model included 9 significant factors
(BNPP, pH, Clay, MAT, MAP, TN, MN, MC, CN), as well as the corresponding high-
spatial-resolution raster datasets (Figure S2–S4). The model performed well, with an
adjusted coefficient of determination ($R^2$) of 0.85, 0.86, and 0.90 for cropland,
grassland, and forestland, respectively, and the RMSE values are all less than 0.01
(Figure 3a-c). The predictions and measurements of all samples were also distributed
close to the 1:1 line. These validations suggest that the trained RF model is capable of
capturing and predicting the spatial pattern of the soil β value on a global scale.

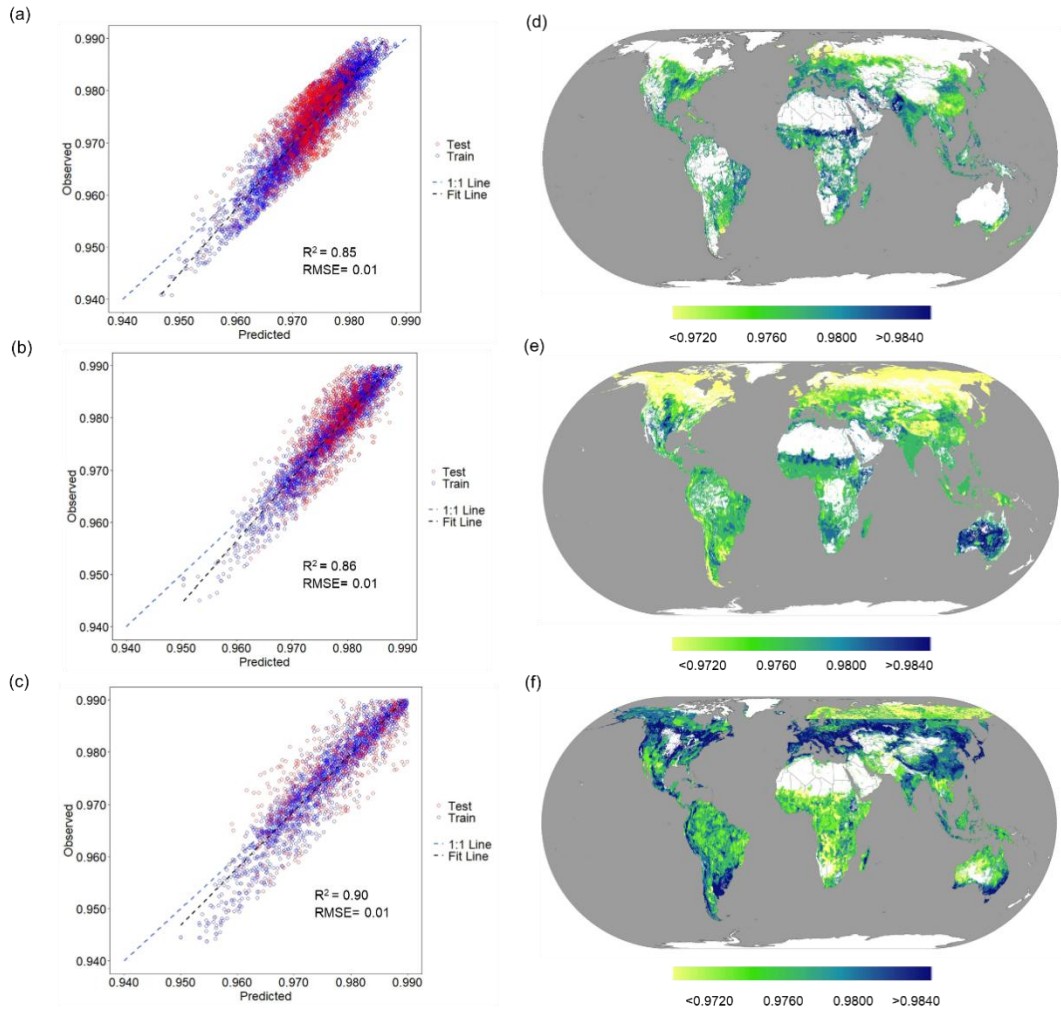


**Figure 3.** Grid-level maps showing the predicted global soil β values. **a–c** reflect the
performance of the random forest model as evaluated by the correlation between the
observed and predicted responses of soil β values. **d–f** illustrate the predicted spatial
variability of predicted soil β values in cropland, grassland, and forestland, respectively.
*3.4 Mapping the global grid-level soil β value*
We predicted the global soil β value using the RF model for 4,057,524 integrated grid-
level, high-spatial-resolution soil and climate raster datasets (cropland, $n$ = 832,827;
forestland, $n$ = 1,695,053; and grassland, $n$ = 1,529,644). The average values were
0.9716 (95% CI: 0.9692-0.9738), 0.9762 (95% CI: 0.9656-0.9831), and 0.9792 (95%
CI: 0.9687-0.9877) for cropland, grassland, and forestland, respectively, with CVs of
4.73%, 1.79%, and 1.94% (Figure 3d-f). The spatial distribution of soil β values across
cropland, grassland, and forest ecosystems reveals both commonalities and notable
differences. High β values are predominantly distributed in tropical and subtropical
regions, including parts of South America, Oceania, and sub-Saharan Africa, whereas
low β values are mainly concentrated in temperate regions, particularly in northern and
western Europe and eastern and northern North America. Notably, the distribution of
high β values varies across ecosystems. High β values are primarily observed in sub-
Saharan Africa, central North America, and southern Oceania in cropland (Figure 3d).
For grassland, mainly concentrated in southeastern South America, southern Africa,
and Oceania (Figure 3e). Forestland exhibited the most extensive distribution of high β
values, spanning southern South America, central and southern Africa, and Oceania
(excluding the central region) (Figure 3f). Cropland exhibits a more confined range of
low values, mainly in northwestern Europe, while grassland and forestland display
broader areas of low values, particularly across eastern and northern North America.
These patterns underscore the geographic variability of soil β values, reflecting the
complex interplay between environmental and ecological factors shaping these spatial
distributions.
*3.5 Spatial variability of the SOC density in subsoil*
The estimated values for the global average SOC density of cropland, grassland, and
forestland 62 Mg ha$^{-1}$ (95% CI:52-73), 70 Mg ha$^{-1}$ (95% CI:57-83), and 97 Mg ha$^{-1}$
(95% CI:80-117), respectively, for the 20–100 cm layer (Table S1), with considerable
spatial variation on the global scale (Figure 4). The larger the soil β value, the more
rapidly the SOC density decreased with an increase in soil depth. Spatially, there was
geographic variability in the SOC density depending on ecosystems. The higher values
exhibited similar spatial patterns in each ecosystems type and were distributed mainly
in northern and western Europe and northern and eastern North America.
For cropland, lower SOC density values were predominantly distributed in Eastern and
Southwestern Asia, Sub-Saharan Africa, Southern Africa, Central North America, and
Southern Oceania. In contrast, higher SOC density values were mainly concentrated in
temperate regions, such as parts of Europe, Northern North America, and some regions
in South America (Figure 4a). For grassland, SOC density showed significant spatial
variation, with lower values primarily distributed in Eastern and Southwestern Asia,
Eastern and Southern South America, and Oceania. In contrast, higher values were
concentrated in temperate regions, such as Northern and Western Europe, Northern
North America (Figure 4b). For forestland, SOC density displayed clear spatial
heterogeneity. Lower values were primarily distributed in Northern South America,
Central and Southern Africa, Northeastern Africa, and the Central region of Oceania,
areas often characterized by tropical or subtropical climates with rapid organic matter
decomposition rates (Figure 4c). In contrast, higher values were predominantly found
in temperate and boreal forest regions, including northern and Western Europe,
Northern North America, and parts of Eastern Asia. The spatial variation in SOC
density at multiple depths (20–40, 40–60, 60–80, and 80–100 cm) was also estimated
(Figure S5–S7), which exhibited a decreasing trend with increasing depth.

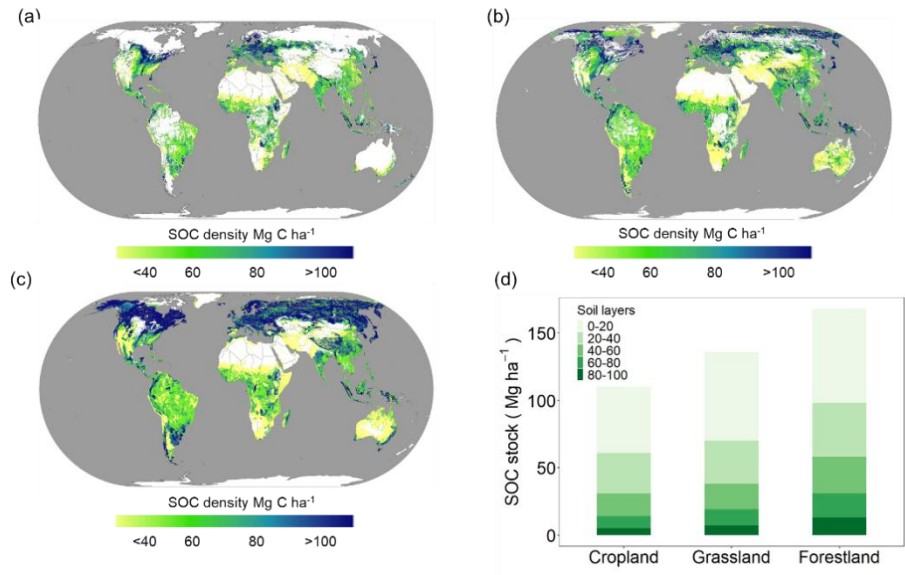


**Figure 4.** Grid-level maps showing the predicted global subsoil SOC density for the
20–100 cm soil layer. **a–c** represents cropland, grassland, and forestland, respectively.
The Plot d shows the SOC density in soil profiles of cropland, grassland, and forestland.
***3.6 Uncertainty analysis of subsoil SOC density across ecosystems***
Overall, regions with high uncertainty are concentrated in tropical and subtropical areas,

such as sub-Saharan Africa, Southeast Asia, the Amazon region of South America, and parts of Oceania. In contrast, regions with low uncertainty are primarily located in temperate and boreal areas, including northern Europe, Northern North America, and Northern Asia. Among them, forestland exhibits slightly higher SOC density prediction uncertainty (38%) compared to grassland (37%) and cropland (34%) (Figure 5).

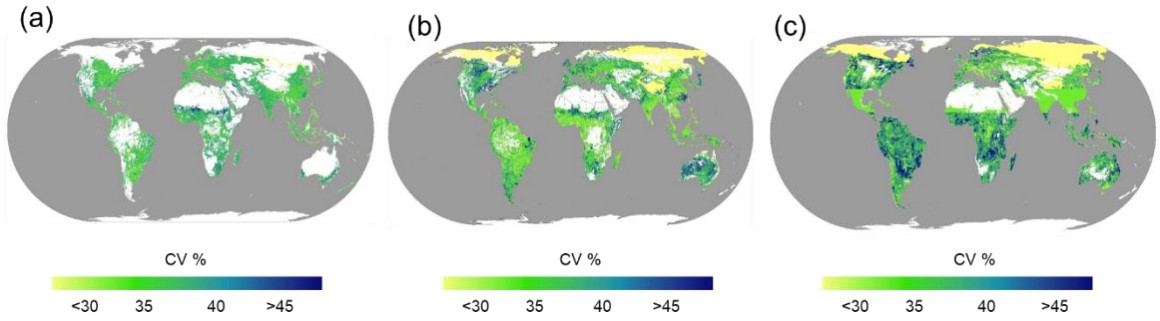

**Figure 5.** Grid-level maps illustrating the uncertainty of predicted global subsoil SOC density. **a–c** represents cropland, grassland, and forestland, respectively.

**4. Discussion**

*4.1 Comparison of high-resolution SOC dynamics*

Global estimations of SOC stock reported in the literature exhibit considerable variation. The estimated SOC stocks for cropland, grassland, and forestland (Table 1) in our study align closely with previous studies (Liu et al., 2021; Conant, 2010; Dixon et al., 1994). The SOC stock of all land in the 0–100 cm soil layer was 1418 Pg (95% CI:1276-1577), which was slightly lower than the estimate reported by Sanderman et al. (2017) and Batjes. (1996). However, we believe that our estimation was not underestimated. This discrepancy may be due to the overestimation in (Sanderman et al., 2017), which could be attributed to the suboptimal quality of the training dataset used in their spatial prediction models ($R^2$=0.54). Earlier assessments (Batjes, 1996) relied on databases that included very few soil profiles from regions such as North America, Oceania, or the northern temperate zones. The subsoil SOC stock of all land was 803 Pg (95% CI:661-962), which was consistent with other research results (Scharlemann et al., 2014; Roland Hiederer. and Köchy., 2011; Zhou et al, 2024). We found that the subsoil contains 57% of total SOC stock in the top 0-1 m soil layer, which is consistent with the percentages cited in previous works (47-55%) (Lal, 2018; Balesdent et al., 2018). Overall, this demonstrates the feasibility and accuracy of our methodology, with the

estimations proving to be relatively accurate
Similar to the findings of Tao et al. (2023) our study reveals a global SOC density
pattern with lower values at low latitudes and higher values at high latitudes. The
vertical migration of organic matter is notably more pronounced in northern permafrost
regions compared to other areas. For cropland, consistent with the estimates by Wu et
al. (2024) the spatial variation in relative SOC density across China shows higher
carbon densities in the Northeast Plain, the Yangtze River Basin, and the southeastern
hills, while lower values are observed in the arid regions of Northwest China (e.g., the
Taklamakan Desert) and the North China Plain. This pattern aligns well with the trends
identified in our study. The FAO report "Global Assessment of Grassland Soil Carbon:
Current Stocks and Sequestration Potential" aligns with our findings, highlighting high
grassland carbon stocks in central China, Northern Russia, Northern Asia, Southeastern
South America, and Central North America. However, our study also identifies Europe
as having significant carbon stocks. This is mainly because temperate climate,
particularly in Northern and Western Europe, is humid and mild, providing favorable
conditions for the formation and accumulation of soil organic matter. Unlike croplands
and grasslands, forestlands are long-lasting vegetation types, with SOC strongly shaped
by local environmental conditions. Zhang et al. (2024) predicted forest SOC stocks
across climatic zones and soil types, showing higher stocks in Europe, Russia, and
Canada. Mediterranean and temperate regions also have higher SOC than
tropical/subtropical regions, consistent with our findings, though their study only
considers surface soil.
Additionally, we observed higher SOC density in boreal forests and tundra regions,
showing spatial variability consistent with the spatial variation in carbon turnover times
reported in other study (Li et al., 2023), particularly in northern high-latitude permafrost
and tundra areas. This suggests that in low-temperature environments, longer soil
carbon turnover times, and lower microbial activity reduce the decomposition rate of
soil organic matter, allowing more SOC to accumulate. The highest SOC density and
microbial C/N ratios were found at high latitudes in tundra and boreal forests, probably
due to the higher levels of organic matter in soils, greater fungal abundance, and lower
nutrient availability in cold biomes (Gao et al., 2022).
Our estimated SOC density at 111 Mg ha$^{-1}$ (95% CI:101-122) for cropland (Table S1)
was higher than that reported in other study (Liu et al., 2021), and lower than that of
tropical cropland (Reichenbach et al., 2023). For forestland, the SOC stock was
estimated at 177 Mg ha$^{-1}$ (95% CI: 150–187) for the 0–100 cm soil layer (overall),
consistent with the estimate reported by Dixon et al. (1994), but significantly lower than
those observed in mangroves and tropical forestland (Atwood et al., 2017; Reichenbach
et al., 2023). For grassland, it was 132 Mg ha$^{-1}$ (95% CI:119-145) overall, much higher
than that of (Conant et al., 2017). Finally, on a global scale, the SOC density of all land
for the 0–100 cm soil layer was estimated at 136 Mg ha$^{-1}$ (95% CI: 123–151), which is
significantly higher than the estimate reported by Hiederer & Köchy (2011).

**Table 1.** Comparisons of the estimated SOC stocks with other studies

| | Global area ($10^9$ ha) | Topsoil (Pg) 0–20/30 (cm) | Subsoil (Pg) 20/30–100 (cm) | Total (Pg) 0–100 (cm) | References |
|---|---|---|---|---|---|
| Cropland | | 58 | 69 | 127 | Liu et al., 2021 |
| Cropland | 1.20 | 59 | 74 (95% CI:62-88) | 133 (95% CI:121-146) | This study |
| Forestland | 4.10 | 359 | 787 | 1146 | Dixon et al., 1994 |
| Forestland | 5.64 | 395 | 547(95% CI:451-660) | 942 (95% CI:846-1055) | This study |
| Grassland | | | | 343 | Conant, 2010 |
| Grassland | 2.59 | 161 | 181 (95% CI:148-215) | 342 (95% CI:308-376) | This study |
| All land | | 684–724 | 778–824 | 1462–1548 | Batjes, 1996 |
| All land | | 699 | 718 | 1417 | Roland Hiederer. and Köchy., 2011 |
| All land | | 699 | 716 | 1416 | Scharlemann et al., 2014 |
| All land | | 863 | 961 | 1824 | Sanderman et al., 2017 |
| All | | | | 1360 | Zhou et al, et al., 2024 |
| All land | | 615 | 803 (95% CI:661-962) | 1418 (95% CI:1276-1577) | This study |


SOC: soil organic carbon, 95% CI: refers to the confidence interval
***4.2 Factors affecting soil β values and spatial variation***
MAT was the primary drivers of soil β values, exhibiting a significant positive
correlation. Specifically, with the increase of MAT, the β value increases, and the
decrease of SOC density with depth becomes smaller (Figure 2a). This shows that the
higher the β value, the relatively lower the proportion of the SOC storage in the soil
surface (consistent with previous research Hartley et al., 2021; Melillo et al., 2017). It
is generally accepted that in cold and wet regions, low soil temperatures and/or
anaerobic conditions promote the formation of thick organic horizons and peats,
resulting in the storage of large amounts of SOC (Garcia-Palacios et al., 2021). Tropical
soils have the lowest SOC persistence, while polar/tundra soils and soils dominated by
amorphous minerals exhibit the highest SOC abundance and persistence (von Fromm
et al., 2024). These differences indicate that soil β values are high in low-latitude
regions, such as tropical rainforest areas, and low in high-latitude regions, such as the
tundra, showing a spatial distribution pattern. Climate warming may lead to greater
SOC losses in surface soils compared to deeper layers, especially in high-latitude SOC-
rich systems (Wang et al., 2022). Experimental results of long-term warming show that
soil respiration is sensitive to temperature rise (Xu et al., 2015). It could be driven by
the changes in the temperature dependence for microbial process rates (Karhu et al.,
2014). As field experiments have shown that warming can modify microbial physiology
and resource availability (Poeplau et al., 2017).
We found a significant negative relationship between soil $\beta$ values and MAP. This
suggests that higher precipitation rates are associated with a steeper decrease in SOC
density with increasing depth. This is primarily due to the pronounced positive
correlation between MAP and the surface SOC density (Liu et al., 2023). In wetter
climates where the precipitation exceeds evapotranspiration, there is a strong
relationship between mineral-associated SOC concentration and persistence, due to the
humid soil environments that favor greater root growth and abundance (Heckman et al.,
2023). And, the higher the intensity of precipitation, the more susceptible deep soil
carbon is to loss (Sun et al., 2024).
Additionally, BNPP plays a crucial role in the global land carbon cycle and carbon
balance, as it is a major source of SOC. The increase in BNPP, along with greater root
exudates and changes in microbial activity, may lead to new carbon accumulation
(Zheng et al., 2024), which resulted in a decreasing trend of soil $\beta$ values.
Our results highlight the important role of edaphic properties in explaining variation in
soil $\beta$ values, not just climate and biological factors (Figure S1). The soil CN ratio and
soil clay content both exhibited a similar negative correlation with the $\beta$ value. A higher
soil CN ratio may decelerate the decomposition rate of organic matter, thereby
facilitating an increase in SOC content in warm and arid regions (Spohn et al., 2023),
such that the soil $\beta$ values would trend downward. Under soil CN ratio > 15, warming
significantly enhances the development of root biomass (Bai et al., 2023), this could
induce a corresponding SOC accumulation. Clay fraction of the soil can absorb litter-
derived C and microbial-derived C, promoting the accumulation of organic carbon
(Hicks Pries et al., 2023).
Our results showed that for near-neutral pH soils, the $\beta$ values tend to be stable. In
acidic soils, significant losses of SOC occur because microbial growth is more severely
constrained, leading to a reduced efficiency in the decomposition and utilization of
organic matter by microorganisms (Malik et al., 2018). Salinization and alkalization
impede plant growth, leading to reduced biomass and lower organic matter input into
the soil, causing the soil organic carbon content and organic carbon pool to remain very
low (Li et al., 2023). The harsh conditions of saline-alkaline soils hinder microbial
survival and activity, reducing their efficiency in decomposing and utilizing organic
matter. Soil pH had non-linear relationships with microorganisms, tends to be neutral,
and the abundance of microorganisms is higher (Patoine et al., 2022). The combination
of these factors explains the higher β values observed under extreme acidic or alkaline
conditions. Thus, near-neutral pH soils, may enhance its carbon storage potential by
improving microbial growth efficiency and facilitating the channeling of matrix
components into biomass synthesis.
The effects of TN, MC, MN on soil β values exhibited the same trend, which initially
increased and then decreased. The TN stock in the soil exhibits a significant positive
correlation with the SOC stock (Feng et al.,2018), leading to a reduction in the β value
in nitrogen-enriched soils. MC had positive relationships with the SOC content across
the large spatial scale, because of microbes should be considered not only as a
controlling factor of the consumption of SOC, but also as an influencing factor of the
production of SOC (Tao et al., 2023). Microbial necromass has been identified as a
major contributor to SOC formation across global ecosystems (Wang et al., 2021a).
Evidence from China shows that microbial residues contribute a larger proportion of
SOC in subsoils than in topsoil (Wen et al., 2023). Therefore, in soil profiles with a
high microbial carbon and nitrogen, the soil β value is smaller, indicating a steeper
decrease in SOC density with increasing depth.
*4.3 Challenges and opportunities: Deep soil SOC sequestration*
More and more studies have shown about the necessity to better understand subsoil
SOC dynamics. Biotic controls on SOC cycling become weaker as mineral controls
predominate with depth (Hicks Pries et al., 2023). The topsoil is rich in carbohydrates
and lignin, while the subsoil is rich in protein and lipids, the decrease rate of the ratio
of the microbially derived carbon to plant-derived carbon with SOM content was 23%–
30% slower in the subsoil than in the topsoil (Huang et al., 2023). Warming stimulates
microbial metabolic activity on structurally complex organic carbon, resulting in a
larger loss of subsoil polymeric SOC compared to topsoil (Zosso et al., 2023). However,
long-term experiments may not be long enough to quantify SOC dynamics in subsoil,
large-scale research methods and machine learning are particularly important and
necessary. Based on measured soil profile data and environmental variables, Wang et
al. (2021b) employed machine learning methods to assess SOC stocks and spatial
distribution of subsoil in frozen soil areas in the third pole region. The investigation of
deep soil organic carbon is inherently complex and involves intricate and time-intensive
methodologies. This complexity results in a paucity of research data, which
consequently introduces considerable uncertainties into model-derived predictions. To
avoid under- or overestimation of the SOC stocks of an ecosystem, it is important to
consider the subsoil when formulating sequestration policies for the whole soil profile
(Button et al., 2022), as the "4 per 1000" approach for the top 30 to 40 cm soil layer
provides an incomplete representation of the soil profile (Rumpel et al., 2018). It may
be essential to sample the soil deeper (e.g. 0–100 cm) and incorporate deep soils into
future manipulations, measurements and models.
In addition, researchers had quantified the contribution of optimizing crop
redistribution and improved management, and topsoil carbon sequestration in offsetting
anthropogenic greenhouse gas emissions and climate change (Wang et al., 2022b;
Rodrigues et al., 2021; Yin et al., 2023), the ability and consequence of subsoil SOC
sequestration of crop management remains to be further studied. Conducting global-
scale subsoil SOC dynamics studies will fill the knowledge gap to develop appropriate
soil C sequestration strategies and policies to help the world cope with climate change
and food security (Amelung et al., 2020; Bossio et al., 2020). As such, it is crucial that
future research efforts focus on SOC sequestration efficiency with climate change,
considering the entire soil profile.
*4.4 Strengths and limitations*
Our research establishes a scientific foundation for further study of SOC dynamics,
sequestration, and emissions reduction across soil profiles, offering significant insights
for achieving Sustainable Development Goals (SDGs), notably SDG2 (Zero Hunger),
SDG13      (Climate      Action),      and      SDG15      (Life      on      Land)
(https://www.undp.org/sustainable-development-goals). To our knowledge, this is the
first study to present global high-resolution maps illustrating the spatial distribution of
SOC density within soil profiles, derived from soil β values informed by soil properties
and climatic conditions. We observed pronounced variations in SOC density across

ecosystems, with forestland demonstrating the highest densities, followed by grassland and cropland. However, the observed differences in SOC dynamics across these ecosystems were primarily attributed to the dominant biogeochemical properties of the soils (Reichenbach et al., 2023).

In our analysis, we incorporated a broad spectrum of environmental variables, including climatic factors and soil physicochemical properties, to examine subsoil SOC dynamics across different ecosystems. The variability in SOC density decline across soil profiles with depth in most areas underscores the imperative for refined soil management practices. Enhancing carbon sequestration in deeper soil horizons constitutes a promising avenue for future research. For example, increasing plant diversity and crop diversification has reinforced SOC stocks in subsoil, with this benefit amplifying over time (Lange et al., 2023, Xu et al., 2023). Current research has shed light on certain aspects of subsoil SOC sequestration mechanisms and turnover dynamics (Luo et al., 2019; Li et al., 2023). However, implementing targeted policies, such as incorporating organic materials and biochar, remains essential for enhancing the SOC sequestration potential of deeper soils (Button et al., 2022). These strategies could play a critical role in synergistically enhancing soil fertility and mitigating greenhouse gas emissions.

Some important aspects of SOC stocks were not included in this study. For instance, microbial necromass is a key contributor to SOC accumulation (Zhou et al., 2023). Due to difficulties in obtaining management data for grasslands and forestlands, we did not account for potential management-specific factors on soil $\beta$ value estimations. For example, N fertilizer application, irrigation amount, soil tillage practices, and organic carbon inputs (straw return, crop residues, and litterfall) may influence the vertical movement of SOC. Moreover, organic carbon inputs can alter SOC decomposition rates, particularly in deeper soil layers (Cardinael et al., 2018).

We also acknowledge that soil layers may not always reach 1 meter, especially in mountainous areas. Due to the lack of global soil thickness data, this limitation may lead to overestimation or underestimation of soil carbon storage in some regions. Focusing on 1-meter profiles provides a reasonable approximation of SOC storage across different ecosystems. Although this approach may not fully capture the variation in soil thickness in high mountain areas, it enables us to gain valuable insights into SOC

dynamics within the global carbon cycle. Future studies will incorporate more detailed soil thickness data to improve our understanding of SOC distribution.

**5. Data availability**

The data of "global patterns of soil organic carbon distribution in the 20–100 cm soil profile for different ecosystems: a global meta-analysis" are available at https://doi.org/10.5281/zenodo.14787023 (Wang et al., 2025). The file named "Rawdata.xlsx" contains data sourced from the literature. The file name is "GE_β.tif", GE represents global ecosystems, which including cropland (CL), grassland (GL), and forestland (FL). "FL_β.tif" represents the spatial distribution of β for forestland at 20-100 cm depth. The file name is "GE_d_SOCD.tif", where SOCD represents soil organic carbon density, d represents soil depth, for example, "FL_20-100_SOCD.tif" represents the spatial distribution of SOCD for forestland at 20-100 cm depth.

**6. Conclusion**

Accurately quantifying the distribution of soil profile SOC stocks is crucial for C sequestration and mitigation. Herein, machine learning was applied to the β model to estimate SOC stocks in soil profiles at depths of 20–100 cm. The subsoil SOC density values of cropland, grassland, and forestland were estimated to be 62 Mg ha$^{-1}$ (95% CI:52-73), 70 Mg ha$^{-1}$ (95% CI:57-83), and 97 Mg ha$^{-1}$ (95% CI:80-117), respectively, with significant geographic variability across different ecosystems. Additionally, The global subsoil SOC stock was 803 Pg C (95% CI:661-962) (cropland, grassland, and forestland were 74 Pg C (95% CI:62-88), 181 Pg C (95% CI:148-215), and 547 Pg C (95% CI:451-660), in which an average of 57% resided in the top 0–100 cm of the soil profile. This dataset provides a valuable resource for refining existing Earth system models and enhancing prediction accuracy. Furthermore, it offers critical insights into global SOC dynamics and the spatial variability of SOC within entire soil profiles. Our findings also serve as a valuable reference for decision-makers in developing more effective carbon budget management strategies.

**Author contributions**

The study was completed with cooperation between all authors. ZC and YY conceived and designed the research. HW: conceptualization, investigation, methodology, data curation, visualization, conducted data analysis and wrote original draft. XT:

methodology, data curation, visualization, TC: investigation, data curation,
conceptualization, investigation. ZC, KH, ZW, HG, QM, YW, YC, MZ contributed to
the scientific discussions. ZC and QZ: conceptualization, supervision, funding
acquisition.

**Competing interests.**

The authors declare that they have no conflict of interest.

**Disclaimer. Publisher's note**

Copernicus Publications remains neutral with regard to jurisdictional claims in
published maps and institutional affiliations.

**Acknowledgements.**

This work was financially supported by the PhD Scientific Research and Innovation
Foundation of Sanya Yazhou Bay Science and Technology City (HSPHDSRF-2022-05-
013), and the Hainan Provincial Joint Project of Sanya Yazhou Bay Science and
Technology City (2021JJLH0015), and the National Key Research and Development
Program of China (2021YFD1900901).

**Financial support.**

This work was supported by the PhD Scientific Research and Innovation Foundation
of Sanya Yazhou Bay Science and Technology City (HSPHDSRF-2022-05-013), and
the Hainan Provincial Joint Project of Sanya Yazhou Bay Science and Technology City
(2021JJLH0015), and the National Key Research and Development Program of China
(2021YFD1900901).

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
