# Peer review of "for different ecosystems: A global meta-analysis"

_Earth System Science Data, 2024_

## Author Comment (AC1)

**Reply to RC1:**

Thanks for your detailed and constructive feedback. We appreciate the time and effort you invested in reviewing our manuscript. Each of your comments and suggestions has been carefully considered, and we have carefully addressed revised the manuscript accordingly. Below is a detailed response to each point raised:

1. Line 20: the word 'we' is missing.

Reply: Thank you for pointing this out. The missing word "we" has been added, and the updated sentence is now reflected on Line 21.

2. Line 29: adding the range of soil depth would help strengthen this conclusion.

Reply: Thank you for your suggestion. We have added the range of soil depth to strengthen the conclusion. The revised statement is: "SOC density decreases with increasing depth, ranging from 30 Mg ha$^{-1}$ (95% CI: 26-35) to 5 Mg ha$^{-1}$ (95% CI: 4-7) (at depth intervals of 20-100 cm, in 20 cm increments) for cropland, from 32 Mg ha$^{-1}$ (95% CI: 27-37) to 7 Mg ha$^{-1}$ (95% CI: 5-9) for grassland, and from 40 Mg ha$^{-1}$ (95% CI: 34-46) to 13 Mg ha$^{-1}$ (95% CI: 9-17) for forestland". The updated sentence is now reflected on Line 29-33.

3. Line 33: the word "Global" should be in lower case; Grammer error in "which providing".

Reply: Thank you for your comments. We have corrected the word "Global" to lowercase and revised the phrase "which providing" to "providing." The revised sentence now reads (Line 38): "This study provides information on the vertical distribution and spatial patterns of SOC density at a 10 km resolution across global ecosystems, providing a scientific basis for future studies pertaining to Earth system models."

4. Line 41: space between gas and (GHG) is missing.

5. Line 42: a period after the citation is missing.

Reply 4 and 5: Thank you for pointing that out. We have added the missing space between "gas" and "(GHG)", and added the missing period after the citation. The updated sentence is as follows (Line 45-48): "Organic carbon in soil (SOC) plays a critical role in global C cycling, climate change mitigation, reducing greenhouse gas (GHG) emissions, and the health of ecosystems (Bradford et al., 2016; Lal et al., 2021;

Griscom et al., 2017)."

6. Line 45: grammar error in ", which contributes"

Reply: Thank you for your observation. The amended sentence now appears as (Line 50): "Worldwide, high SOC loss due to crop production and grazing significantly contributes to increasing atmospheric $CO_2$ levels (Beillouin et al., 2023; Lal, 2020; Qin et al., 2023)."

7. Line 108: grammar error in " from"

Reply: Thanks for your insights, The revised text now states (Line 110-112): "Profiles with more than three suitable organic carbon measurements in the first meter were included in the analysis, as they provided sufficient detail to characterize the vertical distribution of SOC."

8. The data collected from the literatures should be published as well for validation purposes and promote boarder application by other researchers.

Reply: Thank you for your insightful suggestion. We have published the data sourced from the literature, which not only facilitates validation by the research community but also encourages its broader dissemination and application in various academic activities.

9. While the authors have done a great job collecting literature data with a well global coverage. However, the density of study sites varies significantly across different regions. Please discuss the limitations of this data collection.

Reply: Thank you for your valuable feedback. We recognize that although our data covers a wide global area, the density of study sites varies significantly by region. In the new revision, we have expanded our dataset by integrating additional WoSIS profiles, including 7,636 soil profiles for cropland, 4,534 soil profiles for forestland, and 4,593 soil profiles for grassland to develop the model (Fig.1), which improve the robustness of the models. Additionally, we have added a discussion of these limitations in the revised manuscript, addressing how this variability may affect the results and suggesting directions for future research to mitigate these issues (Line 554-562).

10. Section 2.2: as the logic flows from previous section to this one, it directs reader to believe that this section explains how the authors calculated SOC density and stock

from the literature. It however seems to estimate gridded SOC stock via predicted soil β in the following section. If the latter is the main focus, consider relocating it to the right place (maybe after 2.5).

Reply: Thank you for your suggestions regarding the structural aspects of the methods section in the manuscript. We recognize that the logical flow in Section 2.2 may have led to misunderstandings, leaving readers with the impression that the section primarily focuses on calculating SOC density and stocks from the literature. In response, we have implemented necessary adjustments.

We have divided Section 2.2 into two parts: the first part concentrates on calculating the SOC density for study sites derived from the literature, which is then used to estimate soil β values for Random Forest modeling. The second part addresses how we utilize the predicted soil β values to estimate the global SOC density and stocks across various ecosystems in gridded formats (as described in Section 2.6, after Section 2.5). These revisions enhance clarity regarding data sources and promote a more coherent logical flow throughout the manuscript.

11. Section 2.3: Clarify whether the soil β values were directly obtained from the studies or calculated using Equations 3 and 4. Typically, soil β is calculated from these equations based on known SOC at different depths in the literatures, rather than the reverse. Clarification on this would be helpful.

Reply: Thank you very much for your valuable feedback. We confirm that the soil β values were calculated soil β values were derived using Equations 2 and 3, based on known SOC density data at various depths obtained from the literature. We have clarified this point in Section 2.3 to avoid any potential misunderstandings.

Additionally, we would like to note that the original Equations 3 and 4 mentioned in the initial submission have now been renumbered as Equations 2 and 3 due to a restructuring of the manuscript for better organization and readability. The methodology and calculations remain unchanged, ensuring consistency with the previous approach. We appreciate your careful review and hope the revisions enhance the clarity of this section.

12. Line 146: awkward wording.

Reply: Thank you for your feedback on the wording in Line 146. We have revised this

section to improve clarity and readability (Line 150-151).

13. Section 2.4: consider moving it after 2.5, creating a more logical sequence: extracting data from literatures -> building model to predict soil β -> preparing spatial data -> estimating SOC stock.

Reply: Thank you for your comment. The spatial data of soil and environmental variables need to be prepared (included 9 significant factors (BNPP, pH, Clay, MAT, MAP, TN, MN, MC, CN), as well as the corresponding high-spatial-resolution raster datasets) before the prediction of spatial soil β value. Therefore, the logical sequence: extracting data from literatures -> preparing spatial data -> building model to predict spatial soil β value -> estimating SOC stock.

14. For 1221 soil profiles in 161 studies, the authors could make use of the variability of SOC in each study to estimate the uncertainty range of this global SOC dataset. Given the high heterogeneity of SOC, adding uncertainty estimates could enhance the value of this dataset. This is just a suggestion for the authors' consideration.

Reply: Thank you for your valuable suggestion. We appreciate the importance of estimating the uncertainty range of this global SOC dataset, especially considering the high heterogeneity of SOC. To address this, we conducted a Monte Carlo simulation to estimate the overall uncertainty in the estimated spatial SOC density. The uncertainty primarily arises from the soil β estimation-related parameters and the Random Forest (RF) model. The input parameters in the RF model were assumed to follow independent normal distributions, with the grid value as the mean and its 5% as the standard deviation. We performed 1,000 random samplings to obtain the interval for each grid using Monte Carlo simulations. The sampled values were then used to run the RF model, predicting the grid-level soil β with 100 bootstraps. Then we use predicted grid-level soil β to recalculated the distribution of SOC density (SOCD) across different ecosystem. Finally, we calculated the mean along with the 2.5% and 97.5% percentiles to establish the 95% prediction interval of SOC density and SOC stocks. We believe this approach enhances the value of our dataset by providing uncertainty estimates.

15. The RF generally performs well across three ecosystems. However, it tends to over-estimate the lower β and under-estimate the higher β. The authors need to reset their model to improve it. If it cannot be resolved, an explanation and discussion of the

potential impacts on predicted SOC, particularly regarding spatial distribution (e.g., even lower soil β in boreal grasslands as seen in Figure 3E), should be provided.

Reply: Thank you for your valuable feedback regarding the performance of the Random Forest (RF) model across the three ecosystems. After incorporating additional the WoSIS profile data, the accuracy of the model has significantly improved, with $R^2$ values exceeding 0.85. The slopes for these regressions are all close to 1, indicating that the bias in the previous model was primarily due to insufficient sampling in certain regions.

16. Figure 3: clarify that the numbers in panels d-f represent predictions to avoid confusion.

Reply: Thank you for pointing this out. We have updated the caption of Figure 3 to explicitly state that the numbers in panels D-F represent predicted values to prevent any confusion.

17. Line 304-306: consider moving this explanation to the discussion section.

Reply: Thank you for your suggestion. We accept this excellent proposal and have reflected it in the main text, the updated sentence is now reflected on Line 470-473.

18. Comparing the estimated SOC stocks with other studies across different ecosystems in terms of total numbers is valuable. Additionally, including comparisons with spatial maps would provide a more comprehensive validation of the dataset.

Reply: Thank you for your suggestion. In response, we have included a discussion of the spatial variability in SOC stocks in Section 4.1 (Comparison of high-resolution SOC dynamics) to provide a more comprehensive validation of the dataset.

19. Line 414-415: awkward wording.

Reply: Thank you for your valuable comments. The updated sentence is as follows (Line 499-502): "The investigation of deep soil organic carbon is inherently complex and involves intricate and time-intensive methodologies. This complexity results in a paucity of research data, which consequently introduces considerable uncertainties into model-derived predictions."

**Reply to RC2:**

Wang et al. collected 1221 soil profiles to quantify the vertical distribution of soil organic carbon at global scale. The topic is important and interesting. However, I believe that there are several substantial concerns before publication.

Reply: Thank you very much for your comments and constructive feedback on our study. Below, we provide detailed responses to each of your comments.

1. In year of 2000, Jobbágy & Jackson (2000) quantified the vertical distribution of soil organic carbon with more than 2700 soil profiles up to 3 m. The current dataset just includes 1221 soil profiles, it is too small! The website had provided a lot of soil profiles at global scale, which may help to the current study (WoSIS Soil Profile Database | ISRIC).

Reply: Thank you for your suggestion. We fully agree that the quantity of data is a critical factor improve the results. In the new revision, we have expanded our dataset by integrating additional WoSIS profiles, including 7,636 soil profiles for cropland, 4,534 soil profiles for forestland, and 4,593 soil profiles for grassland to develop the model (Fig.1), which improve the robustness of the models.

2. Jobbágy & Jackson (2000) had evidenced that the equation 3 had the worst performance in fitting the vertical distribution of soil organic carbon, the author should provide the rationality of the functions used.

Reply: We fully understand your concerns regarding the performance of Equation 3. Jobbágy & Jackson (2000) demonstrated that SOC distribution with soil depth follows both exponential and logarithmic patterns. However, the choice of fitting functions often varies across studies, depending on the dataset characteristics and research objectives. Compare with Equations 1 and 2, Equation 3 offers greater flexibility and could increase the comparability of data derived from different studies (Yang et al., 2011; Li et al., 2012). We chose Equation 3 primarily because it effectively captures SOC distribution trends across different ecosystems and has been widely adopted in previous studies, yielding consistently reliable results (Deng et al., 2014; Liu et al., 2018).

Li D , Niu S , and Luo Y.: Global patterns of the dynamics of soil carbon and nitrogen stocks following afforestation: a meta analysis, New Phytol., 195(1):172-81 https://doi/10.1111/j.1469-8137.2012.04150.x. 2012.

Yang, Y., Luo, Y., and Finzi, A. C.: Carbon and nitrogen dynamics during forest stand development: a global synthesis, New Phytol., 190,977-989, https://doi.org/10.1111/j.1469-8137.2011.03645.x, 2011.

Deng, L., Liu, G. B., and Shangguan, Z. P.: Land-use conversion and changing soil carbon stocks in China's 'Grain-for-Green' Program: a synthesis, Glob. Change Biol., 20, 3544-3556, 10.1111/gcb.12508, 2014.

Liu, X., Yang, T., Wang, Q., Huang, F., Li, L.: Dynamics of soil carbon and nitrogen stocks after afforestation in arid and semi-arid regions: A meta-analysis, Sci. Total Environ., 618, 1658-1664, https://doi.org/10.1016/j.scitotenv.2017.10.009, 2018.

3. The calculation of SOC density by Equation 1 had great limitations due to the faction of gravel content.

Reply: Your suggestions are extremely valuable. We recognize that gravel content may have introduced certain errors in the calculation of SOC density. Therefore, in this revision, we have incorporated gravel content into the calculations, see the methods.

4. The calculation of global SOC storage is based on Equation 2 ($SOC\ density\ *$ area of cropland, grassland or forestland). This calculation of useless because of the small data set and greater uncertainty. I suggest that the author focuses on vertical distribution itself.

Reply: Thank you for your suggestions. Actually, after we incorporated a substantial amount of additional profile data, the uncertainty is greatly reduced, and the accuracy of the results is greatly improved. We have updated this part of the data. But it is undeniable that there is still uncertainty in the revised version, so we have carried out uncertainty analysis, see Figure 5.

5. In 2.2 Global soil attributes calculation, why divide the soil profiles into 5 layers with 20 cm intervals? The vertical distribution of SOC can be quantified by correlation between SOC and depth directly.

Reply: The reason why we dividing the soil into 20 cm layers is mainly that no matter the profile data obtained from literature or WoSIS Soil Profile data, a large proportion was not complete soil bulk density data, thus we need to extract it from third-party data (HWSD) sources to facilitate the calculation of soil carbon storage.

Unfortunately, the only soil bulk density data (soil bulk density in different profiles) we can obtain is divided into five levels. To be consistent with this data source, we have to a divide the soil profiles into 5 layers with 20 cm intervals.

6. The selection of the predictors for β needs clear motivation. For example, how did microbial biomass carbon and nitrogen influence the vertical distribution of β? The β is calculated by SOC, therefore, it is a bad choice including SOC as a predictor. The vertical distribution of SOC should be regulated by root or belowground net primary productivity (Xiao et al. 2023).

Xiao, L., Wang, G., Chang, J., Chen, Y., Guo, X., Mao, X., Wang, M., Zhang, S., Shi, Z., Luo, Y., Cheng, L., Yu, K., Mo, F., and Luo, Z.: Global depth distribution of belowground net primary productivity and its drivers. Glob. Ecol. Biogeogr., 32, 1435–1451. https://doi.org/10.1111/geb.13705, 2023.

Reply: Thank you for your suggestion. We selected microbial biomass carbon and nitrogen as predictors because they are closely related to the decomposition rate and vertical distribution of SOC (Tao et al., 2023; Wang et al., 2021). For the choice of SOC as a predictor, we agree with you, which is not as good as root or belowground net primary productivity. In new version, we have integrated BNPP data into the model development and made the new prediction of soil β.

Tao, F., Huang, Y., Hungate, B. A., Manzoni, S., Frey, S. D., Schmidt, M. W. I., Reichstein, M., Carvalhais, N., Ciais, P., Jiang, L., Lehmann, J., Wang, Y. P., Houlton, B. Z., Ahrens, B., Mishra, U., Hugelius, G., Hocking, T. D., Lu, X., Shi, Z., Viatkin, K., Vargas, R., Yigini, Y., Omuto, C., Malik, A. A., Peralta, G., Cuevas-Corona, R., Di Paolo, L. E., Luotto, I., Liao, C., Liang, Y. S., Saynes, V. S., Huang, X., and Luo, Y.: Microbial carbon use efficiency promotes global soil carbon storage, Nature., 618, 981-985, https://doi.org/10.1038/s41586-023-06042-3, 2023.

Wang, C., Qu, L., Yang, L., Liu, D., Morrissey, E., Miao, R., Liu, Z., Wang, Q., Fang, Y., and Bai, E.: Large-scale importance of microbial carbon use efficiency and necromass to soil organic carbon, Glob. Change Biol., 27, 2039-2048, https://doi.org/10.1111/gcb.15550, 2021a.

7. Figure 3 showed greater bias of the current model in predicting the global pattern of β because the slope is not equal to one. In addition, observed (in the y-axis) vs. predicted (in the x-axis) regressions should be used (Guerschman & Paruelo, 2008). Piñeiro, G., Perelman, S., Guerschman, J. P., & Paruelo, J. M. (2008). How to evaluate models: observed vs. predicted or predicted vs. observed?. *Ecological modelling*, *216*(3-4), 316-322.

Reply: Thank you for your suggestion. After added additional WoSIS profiles data, the accuracy of the model has been greatly improved. The slopes for these regressions are all close to 1, which indicated that bias of previous model was attributed to insufficient sampling in certain regions. In addition, we changed the regressions with observed (in the y-axis) vs. predicted (in the x-axis) in the new manuscript.

I believe that the writing of the current study needs to be improved.

Reply: Thank you. We have conducted a thorough review during the revision process, refining the language and enhancing the logical structure to ensure the clarity and rigor of our research presentation.

**Reply to CC1:**

1. Title: "Global patterns of soil organic carbon dynamics in the 20–100 cm soil profile" used the dynamics was incorrect. Dynamics are usually changes on a time scale, where distribution or variation is more appropriate.

Reply: Thank you for your suggestion. We have revised the title with "Global patterns of soil organic carbon distribution in the 20–100 cm soil profile for different ecosystems: A global meta-analysis ".

2. In the abstract, what is soil beta, I think it should be given an explanation.

Reply: Thank you for your suggestion. β is the relative rate of decrease in the SOC density with soil depth (Line 19). We have added it into the abstract.

3. The main innovation of this paper is the accuracy of soil beta value. However, soil bulk density (BD) is an important factor in evaluating the accuracy of soil carbon storage. In this paper, only used Shangguan's (2014) empirical equation to predict the missing value of BD, so the spatial distribution of soil BD is not accurate in this study. Like that of soil beta, which brings uncertainty to the evaluation.

Reply: Thank you very much for your valuable comment. Actually, Shangguan's (2014) provides a database with a more robust representation of BD from 0-1m. We use the database replenished the missing value of BD, rather than relying on empirical equations. In addition, after further revision, we have replaced the BD data source with the Harmonized World Soils Database version 2.0 (HWSD v2.0, https://gaez.fao.org/pages/hwsd), which provides BD datasets for the entire soil profile at a resolution of 1 km.

4. In addition, in order to accurately evaluate the spatial distribution of soil organic carbon, in addition to accurate SOC and BD, accurate soil thickness is also an important parameter. Because not all areas of the soil layer thickness can reach 1 m soil layer, especially in the high mountains.

Reply: Yes, you are right. That is a vital point. We acknowledge that in some mountainous regions, soil thickness may be less than 1 meter. However, we are unable to acquire data on global soil thickness, which suggests that our current assessment results may either overestimate or underestimate carbon storage in the soil. Accordingly, we have incorporated a discussion on this uncertainty in our analysis. Focusing on these

1-meter profiles provides a reasonable approximation of SOC storage across various ecosystems. While this approach may not capture all the nuances of soil thickness variability in high mountain areas, it enables us to generate valuable insights into SOC dynamics in the context of global carbon cycling. In future studies, we will consider a more detailed analysis of soil thickness variability to further enhance our understanding of SOC distribution.

5. How to use machine learning method to accurately establish the spatial distribution map of global soil organic carbon needs to be explained in detail in the research method.
Reply: Thank you. In our research methodology, we present a comprehensive explanation of how to accurately establish the global spatial distribution map of soil organic carbon (SOC) using machine learning techniques. We began with an extensive review of the relevant literature from 1980 to 2022, focusing on SOC stocks and concentrations in soil profiles from croplands, grasslands, and forest ecosystems. This review facilitated the construction of a robust database that incorporates sampling depths along with key environmental factors, including Belowground net primary productivity (BNPP), CN ratio, microbial biomass carbon (MC), microbial biomass nitrogen (MN), soil Clay, soil pH, mean annual temperature (MAT), and mean annual precipitation (MAP).

Building upon this database, we conducted redundancy analysis to identify significant environmental variables affecting soil $\beta$ values. We then established a Random Forest (RF) model to predict and estimate grid-level soil $\beta$ values across different ecosystems. Prior to modeling, we utilized the bootstrap sampling method implemented in the "e1071" R package to determine the optimal parameter values for mtry and ntrees. The predictive capability of the model was rigorously validated through 10-fold cross-validation, with 70% of the data allocated for model training and 30% for validation. Model performance was assessed using the coefficient of determination ($R^2$) and root mean square error (RMSE), achieving $R^2$ values of 0.85, 0.86, and 0.90 for cropland, grassland, and forestland ecosystems, respectively, indicating strong predictive performance.

Through these methods, we successfully established high-resolution (10 km $\times$ 10 km grid) spatial distribution maps of soil $\beta$ values. Subsequently, leveraging the relationship between soil $\beta$ values and soil organic carbon density, we computed the global high-resolution (10 km $\times$ 10 km grid) spatial distribution map of soil organic

carbon density. This integrative strategy combines traditional depth functions with machine learning methodologies, significantly enhancing the predictive accuracy of SOC distribution in soil profiles and providing essential data support for future ecosystem management and carbon budgeting.

6. It is not meaningful to calculate the global average soil carbon density of forestland, grassland and farmland because soil carbon density is spatially very heterogeneous.

Reply: Thank you for your insightful comment. We recognize that soil carbon density is spatially heterogeneous, which presents challenges in calculating a global average for forestland, grassland, and farmland. We actually calculated the spatially very heterogeneous in this study. It is just expressed as an average. To address this concern, we conducted a further uncertainty analysis in our study, which provided a more accurate representation of the research range. Specifically, the 95% confidence intervals for soil carbon density in the 20-100 cm profiles are as follows:

Cropland profiles: 62(95% CI:52-73) Mg C ha$^{-1}$

Grassland profiles: 70(95% CI:57-83) Mg C ha$^{-1}$

Forestland profiles: 97(95% CI:80-117) Mg C ha$^{-1}$

By incorporating these confidence intervals, we aim to acknowledge and account for the spatial variability in soil carbon density, thus improving the robustness of our findings.

7. In addition to forest, grassland and farmland, there are also wetlands and deserts in the terrestrial ecosystem, which are not considered in this paper.

Reply: Thank you for your valuable feedback. We acknowledge that this study focuses primarily on forests, grasslands, and farmlands, and does not include wetlands and deserts in the terrestrial ecosystem. This decision was made due to the lack of sufficient data, particularly regarding soil thickness data that reaches a depth of 1 meter for these ecosystems. However, we recognize the importance of wetlands and deserts in global carbon dynamics and will consider including them in future studies to provide a more comprehensive assessment of soil organic carbon across different terrestrial ecosystems.

---

## Referee Report (RR1)

The estimation of deep soil carbon using soil β is a well-established methodology. While the authors' global spatial prediction of soil β demonstrates scientific merit, the following methodological and analytical issues require explicit clarification and resolution prior to publication.

1) The abstract is too lengthy and should be condensed.

2) In line 23, the claim of 17,984 soil profiles appears inconsistent with the Rawdata.xlsx file containing only 6,817 observational records. This discrepancy requires explicit clarification and public disclosure of the methodological framework governing profile identification and data aggregation protocols. Without clarification of this discrepancy and explicit documentation of the data sources and methodological transparency regarding profile identification criteria, I cannot recommend this manuscript for publication.

3) Lines 96-97: The manuscript states that 17,984 soil profiles were sourced from 14,535 sites. This raises a critical methodological question: Can multiple soil profiles coexist at a single geographic site? If such spatial clustering of profiles exists, it is imperative to clarify the criteria for profile differentiation (e.g., vertical/horizontal sampling intervals, land-use stratification, or temporal replication) and ensure these distinctions are systematically annotated in the publicly available Excel dataset.

4) Line 104: Until which month in 2022 was the literature search conducted?

5) Lines 106-107: Please search and compare the results for: 1. "Soil organic carbon" AND "subsoil" AND "Soil profile"; 2. "Soil organic carbon" AND "Deep soil" AND "Soil profile"; 3. "Soil organic carbon" AND "Soil profile".

6) Line 210: What is the rationale for the standard deviation being presented as 10% of the mean?

7) Figure 1 requires modification. It is recommended to relocate Figure S1 to the main text, combine it with the existing Figure 1 as a panel figure, and select several representative soil profiles to graphically demonstrate the variations in soil β across different ecosystem types.

8) The Random Forest modeling data, including response variables and predictor variables, should be made publicly accessible. The critical code is also recommended to be publicly available.

---

## Author Response (AR2)

**Response to RC1:**

The estimation of deep soil carbon using soil β is a well-established methodology. While the authors' global spatial prediction of soil β demonstrates scientific merit, the following methodological and analytical issues require explicit clarification and resolution prior to publication.

**Response**: Thank you for your constructive feedback. We appreciate your recognition of the scientific merit of our study. We have addressed the methodological and analytical issues you raised and have provided the necessary explanations and clarifications in the revised manuscript. Below are our detailed responses to each of your comments.

1. The abstract is too lengthy and should be condensed.

**Response**: Thank you for your comment. We have revised the abstract to make it more concise while retaining the key points of our study.

2. In line 23, the claim of 17,984 soil profiles appears inconsistent with the Rawdata.xlsx file containing only 6,817 observational records. This discrepancy requires explicit clarification and public disclosure of the methodological framework governing profile identification and data aggregation protocols. Without clarification of this discrepancy and explicit documentation of the data sources and methodological transparency regarding profile identification criteria, I cannot recommend this manuscript for publication.

**Response**: Thank you for your valuable feedback. The discrepancy between the 17,984 soil profiles and the 6,817 observational records in the Rawdata.xlsx file arises because the Rawdata.xlsx file contains original data extracted from the literature, while the remaining profiles come from the publicly accessible WoSIS Soil Profile Database (https://www.isric.org/explore/wosis). We have already clarified this in the manuscript and will further emphasize the profile identification and data aggregation methodology in the revised version to ensure methodological transparency.

3. Lines 96-97: The manuscript states that 17,984 soil profiles were sourced from 14,535 sites. This raises a critical methodological question: Can multiple soil profiles coexist at a single geographic site? If such spatial clustering of profiles exists, it is imperative to clarify the criteria for profile differentiation (e.g., vertical/horizontal

sampling intervals, land-use stratification, or temporal replication) and ensure these distinctions are systematically annotated in the publicly available Excel dataset.

**Response**: Thank you very much for your insightful comment. After careful verification and review, we confirm that multiple soil profiles can indeed coexist at a single geographic site. The criteria for profile differentiation primarily stem from differences in sampling based on fertilizer treatments, crop cultivation systems, tree age, species, sampling time, vertical sampling intervals, and soil profiles ID. We have made sure to clearly explain and organize this information in the publicly available dataset. These distinctions have been systematically annotated to ensure clarity. We hope this addresses your concern.

4. Line 104: Until which month in 2022 was the literature search conducted?

**Response**: Thank you for your question. After careful verification, our literature search was conducted until January 2023.

5. Lines 106-107: Please search and compare the results for: 1. "Soil organic carbon" AND "subsoil" AND "Soil profile"; 2. "Soil organic carbon" AND "Deep soil" AND "Soil profile"; 3. "Soil organic carbon" AND "Soil profile".

**Response**: Thank you for your valuable suggestions and comments on our paper. In response to the retrieval strategy and results you mentioned, we conducted a detailed analysis and comparison, as outlined below:

Search terms and result comparison:

Term 1: Using the keywords "Soil organic carbon" AND "subsoil" AND "Soil profile", we retrieved 818 relevant results.

Term 2: Using the keywords "Soil organic carbon" "Deep soil" and "Soil profile", we retrieved 2,038 relevant results.

Term 3: Using only the keywords "Soil organic carbon" and "Soil profile," we retrieved 13,972 relevant results.

From the above results, it can be seen that adding "subsoil" or "Deep soil" as search terms significantly reduced the number of retrieved results. However, due to our strict literature selection criteria, which only include studies with more than three measurements of organic carbon in the first meter of the soil profile, the number of relevant articles meeting these criteria is relatively low and similar to the amount we have currently obtained. Additionally, after careful verification, we identified a

language expression error in the original search term. The correct term should be: "Soil organic carbon" AND "Soil profile" OR "Subsoil" OR "Deep soil". Thank you again for your careful observation and feedback.

6. Line 210: What is the rationale for the standard deviation being presented as 10% of the mean? Line 210:

**Response**: Thank you for your question. In Monte Carlo simulations, to reduce computational load and improve simulation efficiency, we simplify the distribution of input parameters. The choice to set the standard deviation as 10% of the mean is based on common assumptions to reflect the uncertainty of the soil $\beta$ value estimation and the input parameters in the Random Forest (RF) model (Liu et al., 2024;Xu et al., 2023; Vande et al., 2004).

Liu, Y., Zhuang, M., Liang, X., Lam, S. K., Chen, D., Malik, A., Li, M., Lenzen, M., Zhang, L., Zhang, R., Zhang, L., and Hao, Y.: Localized nitrogen management strategies can halve fertilizer use in Chinese staple crop production, Nat. Food, https://doi.org/10.1038/s43016-024-01057-z, 2024.

Xu, Y., Xu, X., Li, J., Guo, X., Gong, H., Ouyang, Z., Zhang, L., and Mathijs, E.: Excessive synthetic fertilizers elevate greenhouse gas emissions of smallholder-scale staple grain production in China, *J. Cleaner Prod.*, https://doi.org/10.1016/j.jclepro.2024.128671, 2023.

Vanden B., A. J., Gregorich, E. G., Angers, D. A., & Stoklas, U. F. Uncertainty analysis of soil organic carbon stock change in Canadian cropland from 1991 to 2001. Glob. Chang. Biol.., 10(7), 983-994. https://doi.org/10.1111/j.1365-2486.2004.00780, 2004.

7. Figure 1 requires modification. It is recommended to relocate Figure S1 to the main text, combine it with the existing Figure 1 as a panel figure, and select several representative soil profiles to graphically demonstrate the variations in soil $\beta$ across different ecosystem types.

**Response**: Thank you for the great suggestion. We have moved Figure S1 to the main text and merged it with the existing Figure 1 into a panel. We analyzed the variations in soil $\beta$ values under different soil textures. Soil $\beta$ values exhibited significant differences among sandy soil, loam, clay loam, and clay soil. Cropland and grassland ecosystems

exhibited the highest β values in sandy soil, while forest ecosystems showed the highest β values in clay soil.

8. The Random Forest modeling data, including response variables and predictor variables, should be made publicly accessible. The critical code is also recommended to be publicly available.

**Response**: We have uploaded the dataset and key code to a publicly accessible repository are in the process of making the dataset and code available. The data and code can be accessed at https://doi.org/10.5281/zenodo.15019078.